# Physical Activity Level and Quality of Life of Children Treated for Malignancy, Depending on Their Place of Residence: Poland vs. the Czech Republic: An Observational Study

**DOI:** 10.3390/cancers15194695

**Published:** 2023-09-23

**Authors:** Aleksandra Kowaluk, Katarzyna Siewierska, Marie Choniawkova, Petr Sedlacek, Krzysztof Kałwak, Iwona Malicka

**Affiliations:** 1Department of Physiotherapy, Wroclaw University of Health and Sport Sciences, 51-612 Wroclaw, Poland; katarzyna.siewierska@awf.wroc.pl (K.S.); iwona.malicka@awf.wroc.pl (I.M.); 2Supraregional Center of Paediatric Oncology “Cape of Hope”, Wroclaw University Clinical Hospital, 50-556 Wroclaw, Poland; 3Department of Paediatric Haematology and Oncology, University Hospital Motol, 150 06 Prague, Czech Republic; marie.choniawkova@centrum.cz (M.C.); petr.sedlacek@fnmotol.cz (P.S.); 4Department of Pediatric Hematology, Oncology and BMT, Wroclaw Medical University, 50-556 Wroclaw, Poland; krzysztof.kalwak@gmail.com

**Keywords:** oncology, children, paediatrics patients, physical activity, quality of life

## Abstract

**Simple Summary:**

Physical and cardiorespiratory fitness deficits acquired during childhood often persist throughout adult life. It is crucial to understand the reasons for their development during childhood cancer and to prevent them in the early treatment stages. Children treated for malignancy exhibit a reduced level of physical activity (PA); however, it varies depending on their place of residence. In conducted studies, the values of examined parameters were significantly lower in the group of Polish children treated for cancer compared to Czech children. It was also observed that a lower level of PA affected patients’ mental states. Children undergoing cancer treatment who declared sedentary behavior rated their health worse, while children engaging in more frequent PA (lasting at least 60 min daily) felt better mentally, reported higher physical performance, and experienced fewer symptoms of depression. Adequate PA levels have an impact on improving physical fitness, mental well-being, and preventing diseases in civilization.

**Abstract:**

This study aimed to assess the level of physical activity (PA) and quality of life of cancer-treated children, depending on their place of residence (Poland vs. the Czech Republic, where incidence and mortality rates of childhood malignancies are similar). A total of 68 school-age children (7–18 years) undergoing oncological treatment were included in this study. This study used the quality of life questionnaire (KIDSCREEN-10) and the HBSC questionnaire. This study showed statistically significant differences in the level of PA between Polish and Czech children. In Poland, 93.75% of children exhibited no weekly physical effort at the level of moderate to vigorous PA. In the Czech Republic, 69.44% of children engaged in PA lasting at least 60 min per day, or at least 1 day weekly. Physically active children engaging in more frequent effort, at least 60 min daily, reported higher physical performance (rho = 0.41), higher energy levels (rho = 0.41), and less mood disturbance (rho = −0.31). Children with good relationships with parents were more likely to engage in submaximal PA and spend less time on stationary games. Our study showed that an appropriate level of PA improves well-being and quality of life. It is crucial to promote attractive PA programs tailored for cancer-treated children.

## 1. Introduction

Childhood cancers are significant diseases in paediatrics. Every year, approximately 300,000 new cancer cases are diagnosed worldwide among children and adolescents aged 0 to 19 years [1]. Currently, the 5-year survival rate in highly developed countries is more than 80% for cured children [2]. Poland and the Czech Republic have similar levels of socioeconomic development. On a perpopulation basis, similar incidence, mortality, and survival rates for childhood malignancies are observed [3]. Every child, both healthy and undergoing cancer treatment, requires an appropriate amount of daily physical activity (PA). Current WHO (World Health Organization) guidelines for physical activity and sedentary behavior recommend at least 60 min of moderate to vigorous physical activity (MVPA) a day and strength training three times a week [4]. Engaging in physical activity is important and necessary for proper motor and mental development [5].

Currently, children and adolescents exhibit increasingly sedentary behavior and spend more time in front of a screen [6,7]. The level of physical activity in a society is dependent on many factors. These include state politics, the social and cultural environment, environmental diversity, and the climate zone, among others. Available data indicates that a high percentage of children and adolescents, regardless of where they live, do not meet current PA guidelines [8,9].

An increase in sedentary habits is also evident in the group of children being treated for cancer. This is influenced by procedures related to the treatment process (prolonged hospitalizations, frequent chemotherapy, radiotherapy, surgical treatment, adverse and frequent late aftermaths of treatment, i.e., decrease in blood morphotic parameters, malaise, complications of treatment) and excessive sedentary habits of children before treatment as well [10]. Increasingly important is the spread of multimedia technologies and the digitization of society, which causes children to engage less in any activity other than the virtual world. Often, these modern technologies are more attractive, and children prefer them [11].

An international study conducted in 25 countries (including the Czech Republic and Poland) found that the Czech children exhibited the highest levels of PA. At least 60% of Czech children spent >2 h a day on active play. By comparison, in Poland, less than 25% of children devote that much time to active play. In the Czech Republic, nearly all children (98%) were actively spending time—at least for 1 h daily—and 60% of children spent over 2 h daily. In Poland, the respective figures are 82% and 25%. On the other hand, a way to school requiring physical activity (on foot or by bicycle) is reported by 44.5% of young Czechs and 39% of young Poles. A comparable number of children in the Czech Republic and Poland spend less than 2 h a day in front of a screen (64% of children and adolescents in the Czech Republic and 59% in Poland, respectively) [11].

Children undergoing treatment for cancer should also engage in physical activity and should not be deprived of opportunities for exercise. Unfortunately, there are no adequate recommendations regarding the level of PA and sedentary behavior for chronically ill and treated children [12].

Observations made by other researchers confirm that children undergoing treatment for cancer exhibit higher levels of fatigue, physical inactivity, and a decline in their quality of life [13]. In a study by Ho et al., researchers observed that an adequate level of PA during cancer is an important factor affecting the determinants of quality of life in a group of treated patients. In contrast, deficits acquired in childhood also persist long after cancer treatment [14]. 

The diagnosis of cancer limits daily physical activity and affects mental state and, thus, quality of life [15,16]. The aim of our study was to evaluate the level of physical activity and quality of life of children treated for malignancies according to their place of residence: Poland vs. the Czech Republic.

## 2. Materials and Methods

### 2.1. Participants and Recruitment

This study included children undergoing oncological treatment (chemotherapy cycles in hospital settings) from 6 March to 7 April 2023. A total of 68 school-age children (7–18 years, 45 boys and 23 girls) were included. The children were admitted to inpatient treatment at the Clinical Department of Paediatric Bone Marrow Transplantation, Oncology, and Haematology of the Medical University of Wroclaw in Poland (*n* = 32) and at the University Hospital Motol in Prague in the Czech Republic (*n* = 36). Children from Wroclaw and Prague clinics were qualified for the project using a computer program that randomly selected children for the study. Cancer was diagnosed in the case of every participant based on a number of detailed examinations, including cytological evaluation, immunophenotype, genetic, and molecular evaluation. Children were treated according to the appropriate protocol for the given cancer type. The vast majority of children from Poland (Group I) were treated for leukemia and lymphoma (75% of children). The rest of Group I consisted of patients treated for solid tumors—25%. The majority of children from the Czech clinic (Group II), similar to the Polish group, were treated for leukemia and lymphoma (90% of patients). Patients treated for solid tumors accounted for 10% of them. The groups were comparable in terms of the incidence rate, and no statistically significant differences were observed. All children were classified into three risk groups (SR—standard-risk group; IR—intermediate-risk group; HR—high-risk group) based on the following criteria: age, leukocyte count, type of diagnosis, response rate, remission, and cytogenetic results. There were no statistically significant differences between the risk groups. None of the children tested had comorbidities.

The inclusion and exclusion criteria were defined. Inclusion criteria were: diagnosis of malignancy or hematological disease; age between 7 and 18 years; inpatient treatment in the hospital ward; hospital stay > 7 days; and chemotherapy treatment administered. The exclusion criteria were: intellectual disability, previous psychiatric disease, eating disorders, and severe health conditions resulting in contact difficulties.

### 2.2. Research Methods

This study was based on the international quality of life questionnaire related to children’s and adolescents’ health (KIDSCREEN–10) and the HBSC (Health Behaviour in School-aged Children) questionnaire [17]. This study was supplemented with a questionnaire containing basic information about the patient and his/her health condition.

The KIDSCREEN-10 questionnaire is referred to as the mental health index of a child. It can be used as a general instrument to assess the quality of life of children and adolescents suffering from chronic diseases. The aim is to identify children at risk of experiencing a reduced health-related quality of life (their subjective state of health). Responding to the KIDSCREEN-10 index requires only a few minutes. The KIDSCREEN measurement is available in several languages, e.g., Polish and Czech. KIDSCREEN-10 includes four questions about physical and mental health, one question about the relationship with parents, one question about the relationship with peers, two questions about autonomy, and two questions about school.

The physical activity level was assessed using the International Health Behavior Questionnaire HBSC (Health Behavior in School-aged Children) [18]. The survey concerned health-related behaviors and the functioning of children at home, school, and among peers. The questions were related to physical activity in the last seven days. The number of days with at least 60 min of physical activity (MVPA index), the frequency of physical exercise beyond school, and the number of hours spent during leisure time on vigorous physical effort were assessed. The questionnaire also included three questions about the time spent watching TV and using a computer. Paper surveys were used.

### 2.3. Ethics

This study received a positive opinion from the Senate Research Ethics Committee, Wroclaw University of Health and Sport Sciences. Consent number: 7/2018; Date of approval: 8 March 2018.

### 2.4. Statistical Analysis

The statistical analysis was performed in the GraphPad Prism 7 program (Institute of Immunology and Experimental Therapy, Wroclaw, Poland). The normality of the data distribution was assessed using the Shapiro-Wilk test.

In order to assess the statistical significance of result differences between groups (Group I—children from Poland; Group II—children from the Czech Republic) and verify whether groups are comparable, a single-factor analysis was used (with the group as the differentiating factor) to test the differences between groups within a given variable. The post-hoc analysis was conducted using Dunn’s test. No significant differences between the groups were detected.

Spearman’s rank correlation coefficient was used to assess the quality of life of the studied children and the physical activity level in the studied groups. Parameters describing the group characteristics were given by providing descriptive statistics such as the arithmetic mean and standard deviation (SD). The strength of Spearman’s rank correlation was determined as weak correlation < 0.2, low correlation 0.2–0.4, moderate correlation 0.4–0.6, high correlation 0.6–0.8, very high correlation 0.8–0.9, almost complete correlation 0.9–1.0. The significance level was assumed to be *p* < 0.05.

## 3. Results

### 3.1. Participant Characteristics

The height and body weight of every participant were measured prior to this study. Detailed data on age and the parameters above are presented in Table 1.

### 3.2. Physical Health and Mental Well-Being

It was observed that children who were physically active and undertaking more frequent efforts of at least 60 min a day felt better and reported higher physical performance (rho = 0.41), higher energy levels (rho = 0.41), and also less mood disturbance (rho = 0.32) (Table 2). The more often the children made severe physical efforts, the less often they reported feelings of deterioration of physical health and mental well-being (rho = −0.41), as well as lack of energy (rho = −0.34). Children undertaking less submaximal effort were more likely to have a depressed mood and feelings of loneliness (rho = 0.4 and rho = 0.37, respectively) (Table 2).

The level of perceived physical health and mental well-being of the studied children depended on the time spent on computer games or consoles (e.g., PlayStation, Xbox, GameCube), tablets, smartphones, or other equipment, except for motion games. Children who spent more time playing stationary games exhibited deteriorated well-being and physical performance and raised complaints of lack of energy (rho = −0.32, rho = −0.31 during the week, and rho = −0.38, rho = −0.38 at weekends, respectively). The mental state of the studied children also depended on the number of hours spent on stationary games. The more time the children played, the more often they reported decreased mood and a sense of loneliness (rho = 0.32, rho = 0.27 a week, and rho = 0.40, rho = 0.31 at weekends, respectively) (Table 2).

The physical and mental state of the studied children also depended on the use of electronics and the Internet (during the week and at weekends). The more time children spent using a computer, tablet, or phone, the worse they felt in terms of their well-being, physical performance, and level of energy (Table 2).

The time spent sitting in front of the screen (watching movies) determined the mood of the children. Children who spent more time in front of the screen during the week reported a depressed mood and feelings of loneliness (rho = 0.27 and rho = 0.24) (Table 2). The mental state of cancer-treated children also depended on the time allocated to using a computer, tablet, or smartphone for other purposes (homework, sending e-mails, Twitter, Facebook, Instagram, Snapchat, chatting, Internet usage, etc.). Children who used electronic devices more often reported a depressed mood (rho = 0.24) (Table 2).

### 3.3. Autonomy and Independence, as Well as the Relationships with Parents

Children who declared not having enough time for themselves also reported a significantly lower number of hours spent using a computer, tablet, or smartphone for other purposes (homework, sending e-mails, Twitter, Facebook, Instagram, Snapchat, chatting, Internet usage, etc.) both during the week and at weekends (rho = −0.25, rho = −0.31) (Table 3).

Children who reported that they were unable to engage in activities they wanted to perform during the week also spent less time watching movies and TV shows (rho = −0.28). They also spent less time on stationary games during the week as well as at weekends (rho = −0.25, rho = −0.27) and used electronic devices less often during the week (rho = −0.25) (Table 3).

Children in good relationships with their parents more often took submaximal physical efforts. (rho = 0.33), and also spent less time playing stationary games during the week and at weekends (rho = 0.33 and rho = 0.33) (Table 3).

### 3.4. Social Support and the School Environment

Children who spent more time with their peers also had better physical activity scores across all assessed areas. This was reflected in a higher MVPA index, more frequent submaximal physical effort, and fewer hours spent on stationary games, movies, and electronics (Table 4). Children who spent more time watching movies or TV shows, playing stationary games, and using the Internet more frequently at weekends reported problems and worsened ability to focus their attention (rho = −0.32, rho = −0.31, rho = −0.39, rho = −0.34, rho = −0.28) (Table 4).

It was observed that children who were more active and spent less time sitting and less time using the Internet, electronics, and stationary games assessed their health as better. The correlation was observed in all considered aspects, excluding only question HBSC 3.2 (Table 5).

### 3.5. Physical Activity Level–Differences between Groups

A statistically significant difference in the physical activity level (MVPA index level and frequency of effort at submaximal level) was observed between the studied groups. Children from Group II were more often engaged in physical activity of at least 60 min per day than those from Group I (*p* < 0.0001). Children from the Czech Republic were also more likely to make physical efforts at a submaximal level than children from Poland (*p* < 0.0001). The results of this study also showed statistically significant differences between the groups in terms of the number of hours spent playing stationary games and the time spent on electronics during the day. Children treated for malignancy in the Prague clinic spent less time playing games and using electronics (in the week: *p* = 0.0073 and *p* = 0.0009, and at the weekends: *p* = 0.0138 and *p* = 0.0017, respectively) (Table 6).

### 3.6. Quality of Life–Differences between Groups

No statistically significant differences between the groups in the results of the Kidscreen survey on physical health and mental well-being were observed (KID1-KID4) (Table 7).

The results of this study showed no statistically significant differences between the groups in the Kidscreen survey results in the section on independence, relations with parents, and family life (KID5-KID7) (Table 8).

The assessment of the quality of life of the studied children from Poland and the Czech Republic showed a statistically significant difference in the time spent among peers. Group II children spent significantly more time playing and interacting with others (Table 9).

A statistically significant difference was observed in the subjective assessment of one’s own health in a group of studied children. The health self-assessment turned out to be better among the children from Group II (Table 10).

## 4. Discussion 

Children treated for malignancy are the group of patients at particular risk of lack of sufficient physical activity [19,20]. Although they receive high doses of chemotherapy as well as immunosuppressive drugs (after transplantation of allogeneic hematopoietic cells, i.e., ciclosporin, glucocorticoids, or mycophenolate mofetil), they are additionally exposed to the adverse effects of the treatment because of the insufficient level of their everyday physical activity [21,22]. Children with cancer, especially children after hematopoietic stem cell transplantation (HSCT) in transplant wards, require long-term medical supervision and isolation. This is associated with prolonged periods of hospitalization [22]. Long-term social isolation, being in the company of adults only, and the predominance of a sedentary lifestyle negatively affect the patient’s physical performance through limited movement and physical fitness loss [23] and a significant decrease in efficiency parameters [24]. In addition, intensive treatment negatively affects bone mineral density, which is sometimes the cause of osteonecrosis [25].

A frequently observed phenomenon among parents of treated children is a change in parenting strategies. Parents become overprotective of their children and apply a lower level of discipline. This was also confirmed by our study. Of children from Poland, 78.13% could always perform what they wanted in their leisure time, while, in the case of children from the Czech Republic, it was 69.45%. All Polish children and 97.23% of Czech children reported being treated well on par with others. The overprotective behavior of parents is also evident in the case of the food provided to children. Very often, nutrient-poor food becomes an unhealthy encouragement for the child. The priority becomes getting the child to eat more (even unhealthy food or fast food) or using food as a reward for, for example, submitting to a painful procedure [26]. This is an incorrect approach that adversely affects the entire body of the child being treated. Quite often, such habits remain and persist into adulthood. Moreover, these habits, combined with physical inactivity, pose a direct threat and increase the risk of premature death. This issue is increasingly being addressed, with researchers emphasizing the need to study dietary interventions aimed at optimizing the patient’s overall treatment. The goal is to extend survival and reduce the late adverse effects of treatment. The quality of a patient’s diet and the nutrients they take in can potentially affect short- and long-term morbidity and mortality [27].

Loss of both physical and cardiorespiratory fitness negatively affects the child’s participation in daily activities and causes rapid fatigue [28]. The results of our study showed that children with malignancy show insufficient levels of physical activity and do not meet the appropriate level of MVPA. The inadequate levels of the MVPA index and the physical effort frequency at the submaximal level were comparable in both groups. In our study, we observed that children from the Czech Republic and Poland showed insufficient levels of PA; however, Czech children were significantly more physically active and more often took submaximal efforts than Polish children. This observation is analogous to the differences in the level of physical activity undertaken in the group of healthy children [11]. Unfortunately, there are no established standards defining the appropriate level of physical activity for children undergoing cancer treatment. Applying existing norms, which are proper only for healthy children, to the population of cancer-treated children is unreliable. It is necessary to develop specific recommendations for this patient group.

In our study, it was observed that all children from Poland did not meet the recommendations for the minimum level of physical activity according to the MVPA indicator recommended for children. In the case of children from the Czech Republic, 25.01% of them engaged in such moderate activity for 5 or more days a week. Moreover, 93.75% of children from Poland did not engage in physical activity lasting at least 60 min daily for at least 1 day a week, while 69.44% of children from the Czech Republic engaged in continuous physical activity for 60 min at least 1 day a week.

Studies conducted in other countries showed similar results. A prevalence of a sedentary lifestyle and reduced levels of physical activity were observed. Notably, this population remains inactive during treatment, immediately after its completion, and even many years after recovering from the disease. So far, WHO, ACSM (The American College of Sports Medicine), and CDC (Centers for Disease Control and Prevention) have not developed specific recommendations for this population [12].

The quality of life of children with malignancies is closely related to their performance and physical efficiency. Long-term periods of immobilization, social isolation, malaise, and side effects of treatment, i.e., significant pancytopenia, are the other causes of the loss of complete functional independence and the frequent feeling of excessive fatigue [29]. In addition, the quality of life of children undergoing treatment is negatively affected by frequent pain, stress, disturbed family structure, and social stigmatization [30]. In our studies, a link between insufficient physical activity and lower quality of life indicators was observed. Children who were physically active and made more frequent efforts lasting at least 60 min a day were in a better mood, experienced higher physical performance, had more energy, and were less likely to feel depressed.

Probably, children who did not lead a sedentary lifestyle during active cancer treatment are less convinced of their disability. It is crucial for children not to develop a sedentary behavior habit and not to experience prolonged periods of immobilization during treatment. Furthermore, regular physical activity has a beneficial impact on mood and regulates the body’s hormonal balance.

Children who spent more time using electronics and less time on physical activity showed a lowered mood and a sense of loneliness. The relationships between the child and the parents were also important. Children in good relationships with their parents more often made physical efforts at the submaximal level and also spent less time playing stationary games. Some time ago, it was noticed that the disease, especially chronic, does not only concern the patient; its impact should be considered in a broader social context. Malignancy, especially in a child, becomes a specific element of the family structure [31]. Parents can therefore encourage their children, in both good and bad ways, to engage in spontaneous physical activity.

Awareness of the need to maintain a high PA level throughout the course of malignancy treatment remains a crucial element. It is important not only for achieving complete physical performance and independence in adulthood but also for achieving a higher quality of life. From the beginning of treatment, children and their caregivers must be aware of the adverse effects of physical inactivity in everyday life. PA levels must be adequate and suitable for the patient’s needs. The sustainable development of a child with an appropriate level of PA per week is the key to extending life expectancy by improving cardiorespiratory parameters and preventing civilized diseases. Physical and cardiorespiratory fitness deficits acquired in childhood often persist throughout adult life, and it is crucial to know the causes of their development in the course of malignancy and prevent them from occurring during the initial and intensive treatment stage.

Poland, the Czech Republic, Slovakia, and Hungary established an alliance of four Central European countries named the Visegrad Group. These countries have a shared policy and develop similarly. It would seem that the levels of physical activity would be comparable. However, children from Poland scored worse in both groups: the healthy ones [11] and those after oncological treatment. A lower physical activity level influences a mental state. Eventually, it can affect the quality of life in adulthood.

## 5. Strengths and Limitations of This Study

Physical activity level assessment is crucial for the preparation of rehabilitation programs for children treated for malignancy. The implementation and standardization of the rehabilitation programs during the treatment period will affect proper child growth and improve treatment effectiveness by reducing the side effects that result from persistent therapy, consequently improving the quality of life in this group of patients. The novelty of our research is that, so far, no one has examined the level of physical activity and quality of life of children undergoing cancer treatment in Poland or the Czech Republic. So far, our project has examined these values but has not conducted comparative studies with other research centers.

This research restriction could be the size of the tested group, which is too small (n = 68, including 32 children from Poland and 36 children from the Czech Republic). However, the research was performed in a short time (from 6 March to 7 April 2023) to ensure the same performances for all participants, even those resulting from the season and the length of the day. In addition, the group size is also influenced by the disease itself; although there is a steady increase in malignancy, childhood malignancies still account for only 1% of all cases in children. Different diagnoses (leukemia, lymphomas, and solid tumors) were also considered in this study; however, in terms of incidence, the groups were comparable; no statistically significant differences were observed.

## 6. Future Research Directions

It seems necessary to increase the number of studied children, taking into account the type of malignancy and the treatment regimen used. In addition, it is crucial to introduce objective measurements of the physical activity level undertaken, e.g., using pedometers or accelerometers, as well as assess parents’ and caregivers’ psychophysical performance. In the long run, this will allow for assessing the factors determining physical activity practice in families and for verifying possible cultural differences that may have influenced the results.

## 7. Conclusions

Children treated for cancer exhibit reduced levels of physical activity. PA level is different depending on the child’s place of residence. The values of the tested PA level were significantly lower in the group of children from Poland. In our study, it was observed that all children from Poland did not meet the recommendations for the minimum level of physical activity according to the MVPA indicator recommended for children. In the case of children from the Czech Republic, 25.01% of them engaged in such moderate activity for 5 or more days a week. Along with the decrease in physical activity level, there is a decrease in the quality of life. Children who declared a sedentary lifestyle and, at the same time, limited contact with their peers assessed their health as worse. Contacts with peers depended on the children’s place of residence as well. Polish children reported significantly less time spent playing and interacting with their peers.

We observed that, unfortunately, there are no defined standards of PA level for children treated for cancer. Applying existing norms, which are proper only for healthy children, to the population of cancer-treated children is unreliable. It is necessary to develop specific recommendations for this patient group.

It is of utmost importance for health professionals to identify strategies that can help children and adolescents with cancer increase their physical effort and maintain regular activity. This is especially important during the active treatment process. Forms of physical activity that are attractive to the current generation of children, who are accustomed to modern technology, should be sought. Interactive video games or programs using virtual reality could be appropriate.

## Figures and Tables

**Table 1 cancers-15-04695-t001:** Patient characteristics according to sex and group.

	Group I Poland *n =* 32			Group II. The Czech Republic *n =* 36			*p*-Values Group I vs. Group II Dunn’s Test
Determinant Variables	Mean ± SD	Boys Mean ± SD *n* = 22	Girls Mean ± SD *n* = 10	Mean ± SD	Boys Mean ± SD *n* = 23	Girls Mean ± SD *n* = 13	
Age [years]	12.54 ± 3.6	12.91 ± 3.35	11.74 ± 4.12	12.09 ± 3.95	11.68 ± 3.59	12.82 ± 4.58	NS
Height [cm]	151.75 ± 22.86	155.23 ± 24.18	144.1 ± 18.48	146.5 ± 21.78	147.9 ± 22.7	144.04 ± 20.72	NS
Weight [kg]	49.76 ± 22.26	52.34 ± 23.56	44.09 ± 18.93	45.79 ± 23.89	45.3 ± 22.9	46.72 ± 26.45	NS

Note: NS—non-significant.

**Table 2 cancers-15-04695-t002:** Correlation of the Kidscreen results—physical health and mental well-being survey (KID1-KID4), with the overall HBSC results in the whole study group.

Spearman’s Correlation	Well-Being and Physical Performance		Feeling of Energy		Depressed Mood		Feeling of Solitude	
	Spearman’s Rho	*p*-Value	Spearman’s Rho	*p*-Value	Spearman’s Rho	*p*-Value	Spearman’s Rho	*p*-Value
HBSC 1 *	0.41 **	0.00054	0.41 **	0.00062	−0.32 **	0.00733	−0.21	0.09002
HBSC 2 *	−0.41 **	0.00061	−0.34 **	0.00448	0.4 **	0.00072	0.37 **	0.00199
HBSC 3 *	−0.16	0.19829	−0.18	0.14485	0.27 **	0.02406	0.24 **	0.04953
HBSC 3.2 *	−0.12	0.32303	−0.15	0.22457	0.18	0.14580	0.22	0.07820
HBSC 4 *	−0.32 **	0.00766	−0.31 **	0.01104	0.32 **	0.00728	0.27 **	0.02662
HBSC 4.2 *	−0.38 **	0.00126	−0.38 **	0.00142	0.40 **	0.00064	0.31 **	0.01108
HBSC 5 *	−0.23	0.06313	−0.16	0.18753	0.14	0.24864	0.16	0.19529
HBSC 5.2 *	−0.25 **	0.03956	−0.31 **	0.01129	0.24 **	0.04740	0.21	0.08013

Note: * The responses and questions were included in a previous paper [16]. The third question was not considered in the current study since the new HBSC 2018 Questionnaire does not include question 3. ** Results show statistical significance.

**Table 3 cancers-15-04695-t003:** Correlation of results of the Kidscreen survey on independence, parental relationship, and family life (KID5-KID7) with the overall results of the HBSC survey in the whole study group.

Spearman’s Correlation	Having Enough Time for Yourself		The Ability to Take the Actions You Wanted		Fair and Comparable Treatment by Parents	
	Spearman’s Rho	*p*-Value	Spearman’s Rho	*p*-Value	Spearman’s Rho	*p*-Value
HBSC 1 *	0.23	0.05672	0.04	0.76810	−0.35	0.00392
HBSC 2 *	−0.23	0.05935	0.16	0.19998	0.31 **	0.01002
HBSC 3 *	−0.19	0.12235	−0.28 **	0.02274	0.06	0.61176
HBSC 3.2 *	−0.23	0.06288	−0.21	0.08056	−0.1	0.43602
HBSC 4 *	−0.21	0.08748	−0.25 **	0.04316	0.33 **	0.00686
HBSC 4.2 *	−0.19	0.12042	−0.27 **	0.02378	0.33 **	0.00570
HBSC 5 *	−0.25 **	0.03694	−0.18	0.13478	0.11	0.36902
HBSC 5.2 *	−0.31 **	0.01113	−0.25 **	0.03771	0.13	0.27790

Note: * The responses and questions were included in a previous paper [16]. The third question was not considered in the current study since the new HBSC 2018 Questionnaire does not include question 3. ** Results show statistical significance.

**Table 4 cancers-15-04695-t004:** Correlation of the results of the Kidscreen survey on social support by peers and school environment (KID8-KID10) with overall HBSC results in the whole study group.

Spearman’s Correlation	Fun with Peers		Positive Feelings about the School Environment		Ability to Focus Attention	
	Spearman’s Rho	*p*-Value	Spearman’s Rho	*p*-Value	Spearman’s Rho	*p*-Value
HBSC 1 *	0.38 **	0.00135	0.07	0.54862	0.04	0.77207
HBSC 2 *	−0.50 **	0.00001	−0.24 **	0.04846	−0.14	0.25463
HBSC 3 *	−0.27 **	0.02552	−0.25 **	0.04306	−0.32 **	0.00840
HBSC 3.2 *	−0.25 **	0.04079	−0.27 **	0.02469	−0.31 **	0.01001
HBSC 4 *	−0.31 **	0.00979	−0.15	0.23360	−0.39 **	0.00101
HBSC 4.2 *	−0.37 **	0.00183	−0.07	0.57837	−0.34 **	0.00439
HBSC 5 *	−0.36 **	0.00275	−0.19	0.11806	−0.14	0.24454
HBSC 5.2 *	−0.31 **	0.01055	−0.26 **	0.03384	−0.28 **	0.02343

Note: * The responses and questions were included in a previous paper [16]. The third question was not considered in the current study since the new HBSC 2018 Questionnaire does not include question 3. ** Results show statistical significance.

**Table 5 cancers-15-04695-t005:** The correlation of the results of the Kidscreen survey with subjective assessments of patient health (KID 11) with overall HBSC results in the whole study group.

Spearman’s Correlation	Subjective Feelings about One’s Health	
	Spearman’s Rho	*p*-Value
HBSC 1 *	−0.49 **	0.00002
HBSC 2 *	0.55 **	0.0000015
HBSC 3 *	0.31 **	0.01012
HBSC 3.2 *	0.14	0.2536
HBSC 4 *	0.5 **	0.000016
HBSC 4.2 *	0.51 **	0.000011
HBSC 5 *	0.44 **	0.00017
HBSC 5.2 *	0.47 **	0.000058

Note: * The responses and questions were included in a previous paper [16]. The third question was not considered in the current study since the new HBSC 2018 Questionnaire does not include question 3. ** Results show statistical significance.

**Table 6 cancers-15-04695-t006:** Comparison of the percentage results of the survey between the groups on the physical activity level and sedentary behavior (HBSC). The answers in the table are presented in percentages [%].

	HBSC 1 *		HBSC 2 *		HBSC 3 *		HBSC 3.2 *		HBSC 4 *		HBSC 4.2 *		HBSC 5 *		HBSC 5.2 *	
Answer *	Gr. I	Gr. II	Gr. I	Gr. II	Gr. I	Gr. II	Gr. I	Gr. II	Gr. I	Gr. II	Gr. I	Gr. II	Gr. I	Gr. II	Gr. I	Gr. II
1	93.75	30.56	-	8.33	3.13	2.78	3.13	0.00	9.38	13.89	9.38	11.11	6.25	19.44	9.38	16.67
2	6.25	13.89	-	16.67	-	11.11	-	0.00	3.13	13.89	-	8.33	-	25.00	-	22.22
3	-	8.33	3.13	33.33	12.50	30.56	3.13	11.11	6.25	30.56	9.38	19.44	6.25	19.44	6.25	19.44
4	-	11.11	6.25	11.11	18.75	11.11	3.13	22.22	15.63	25.00	6.25	44.44	18.75	2.78	6.25	8.33
5	-	11.11	-	5.56	18.75	1389	28.13	27.78	21.88	8.33	15.63	5.56	21.88	13.89	18.75	11.11
6	-	5.56	6.25	11.11	21.88	16.67	28.13	16.67	25.00	-	31.25	5.56	31.25	8.33	34.38	2.78
7	-	2.78	84.38	13.89	25.00	8.33	25.00	11.11	18.75	8.33	25.00	5.56	15.63	2.78	15.63	8.33
8	-	16.67	-	-	-	-	9.38	2.78	-	-	3.13	-	-	5.56	9.38	5.56
9		-				5.56		8.33		-		-		2.78		5.56
*p*-values	<0.0001 **		<0.0001 **		NS		NS		0.0073 **		0.0138 **		0.0009 **		0.0017 **	

Note: * The responses and questions were included in a previous paper [16]. The third question was not considered in the current study since the new HBSC 2018 Questionnaire does not include question 3. ** Results show statistical significance; NS—non-significant.

**Table 7 cancers-15-04695-t007:** Comparison of the percentage results of the Kidscreen survey between the groups in the section on physical health and mental well-being (KID1-KID4). The answers in the table are presented in percentages [%].

	Feeling Fit and Well KID 1		Feeling of Strength and Energy KID 2		Feeling of Sadness KID 3		Feeling of Loneliness KID 4	
Answer *	Group I	Group II.	Group I	Group II.	Group I	Group II.	Group I	Group II.
1	25.00	11.11	21.88	5.56	3.13	22.22	6.25	63.89
2	40.63	13.89	40.63	27.78	21.88	36.11	46.88	19.44
3	31.25	33.33	25.00	22.22	40.63	30.56	28.13	11.11
4	3.13	25.00	9.38	38.89	21.88	8.33	18.75	2.78
5	-	16.67	3.13	5.56	12.50	2.78	-	2.78
*p*-value	NS		NS		NS		NS	

Note: * The responses and questions were included in a previous paper [16]; NS—non-significant.

**Table 8 cancers-15-04695-t008:** Comparison of the percentage results of the Kidscreen survey between the groups upon the section on independence, parent relationship, and family life (KID5-KID7). The answers in the table are presented in percentages [%].

	Having Enough Time for Yourself KID 5		Performing the Things That You Want to Do KID 6		Being Treated Fairly by Parents/Good Relations with Parents KID 7	
Answer *	Group I	Group II.	Group I	Group II.	Group I	Group II.
1	-	5.56	-	13.89	-	0.00
2	28.13	2.78	21.88	16.67	-	2.78
3	28.13	16.67	31.25	25.00	-	5.56
4	31.25	55.56	25.00	38.89	15.63	36.11
5	12.50	19.44	21.88	5.56	84.38	55.56
*p*-values	NS		NS		NS	

Note: * The responses and questions were included in a previous paper [16]; NS—non-significant.

**Table 9 cancers-15-04695-t009:** Comparison of the percentage results of the Kidscreen survey between the groups in the section on social support by peer and school environment (KID8-KID10). The answers in the table are presented in percentages [%].

	Good Relations with Peers, Fun KID 8		Positive Feelings Related to the School Environment KID 9		Ability to Pay Attention KID 10	
Answer *	Group I	Group II.	Group I	Group II.	Group I	Group II.
1	78.13	19.44	3.13	8.33	-	5.56
2	21.88	13.89	15.63	8.33	28.13	13.89
3	-	13.89	56.25	36.11	25.00	22.22
4	-	25.00	18.75	27.78	37.50	36.11
5	-	27.78	6.25	19.44	9.38	22.22
*p*-values	<0.0001 **		NS		NS	

Note: * The responses and questions were included in a previous paper [16]. ** Results show statistical significance; NS—non-significant.

**Table 10 cancers-15-04695-t010:** Comparison of the percentage results of the Kidscreen survey between the groups upon the section on the subjective assessment of the patient’s health (KID11). The answers in the table are presented in percentages [%].

	Subjective Assessment of Health KID 11	
Answer *	Group I	Group II.
1	-	8.33
2	-	22.22
3	9.38	44.44
4	50.00	16.67
5	40.63	8.33
*p*-values	0.0044 **	

Note: * The responses and questions were included in a previous paper [16]. ** Results show statistical significance.

## Data Availability

The data are available upon reasonable request from the corresponding author.

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
