# Peer review of "Physical Activity Level and Quality of Life of Children Treated for Malignancy, Depending on Their Place of Residence: Poland vs. the Czech Republic: An Observational Study"

_cancers, 2023, doi:10.3390/cancers15194695_

Round 1

Reviewer 1 Report

Do differences in diet in different regions affect cancer treatment?

1.  What is the main question addressed by the research?

The main purpose of the authors is to assess the physical activity level and quality of life of children receiving cancer treatment, depending on their place of residence, the results of the study provide very valuable data for clinical treatment

   2. Do you consider the topic original or relevant in the field? Does it

address a specific gap in the field?

   I think it is original and provides very valuable for clinical treatment in the   field.

   3. What does it add to the subject area compared with other published

material?

   Quality of life and mental status in the place of residence need to be considered in the treatment of childhood malignancies.

   4. What specific improvements should the authors consider regarding the

methodology? What further controls should be considered?

   I would be better to consider whether diet affects the treatment of malignancies, which could be further statistically analyzed.

   5. Are the conclusions consistent with the evidence and arguments presented and do they address the main question posed?

   I think so.

   6. Are the references appropriate?

    No Problem

   7. Please include any additional comments on the tables and figures.

   No additional comments.

Author Response

Thank you very much for reviewing our article. All your comments were very helpful for us and provided add value to make the text better. Please find the answers and our comments below:

The diet is not the main focus of our study. We did not analyse the dietary habits of subjects, so we believe that it is not necessary to address this topic in the text in such way. However, we took note of this suggestion and included a comment on this issue in the Discussion subsection, in lines 365-381. We consider this comment valuable and important for our results. We included this mainly in following sentences:

 ”A frequently observed phenomenon among parents of treated children is a change in parenting strategies. Parents become overprotective of their children and apply a lower level of discipline. This was also confirmed by our study. Children from Poland in 78.13% could always do what they wanted in their leisure time, while, in the case of children from the Czech Republic, it was 69.45%. All Polish children and 97.23% of Czech children reported being treated well on a par with others. The overprotective behaviour of parents also exhibit in the case of the food provided to children. Very of-ten, nutrient-poor food becomes an unhealthy encouragement for the child. The priori-ty becomes getting the child to eat more (even unhealthy food, fast food) or using food as a reward for, for example, submitting to a painful procedure [26]. This is an incorrect approach that adversely affects the entire body of the child being treated. Quite often, such habits remain and persist into adulthood. Moreover, these habits, combined with physical inactivity, pose a direct threat and increase the risk of premature death. This issue is increasingly being addressed, with researchers emphasizing the need to study dietary interventions aimed at optimising the patient's overall treatment. The goal is to extend survival and reduce late adverse effects of treatment. The quality of a patient's diet and the nutrients they take in can potentially affect short- and long-term morbidity and mortality [27].”

Reviewer 2 Report

Paper looks great!

Author Response

Thank you very much for reviewing our article and your kind words regarding it.

Reviewer 3 Report

SUMMARY OF THE STUDY

“Physical activity level and quality of life of children treated for malignancy, depending on their place of residence: Poland vs the Czech Republic”

This study aims to investigate and compare the levels of physical activity and the mental health status of children treated for cancer in Poland vs the Czech Republic.

The paper has an evident social and clinical interest, and I think the document fits the Cancers journal’s aims. The study is interesting, but I think the manuscript cannot be accepted in its present form.

I have detailed my comments and reported some small errors identified below.

GENERAL COMMENTS

·         The summary statement is not focused on the results of the study but provides a general background and the rationale of the study.

·         The abstract is confusing and does not summarize the results.  For example, Lines 29-30: What does the sentence mean? Lines 36-37: Why children from Poland scored worse? An abstract is a short summary of the research paper and should let readers get the gist or essence of the article quickly, preparing them to follow the detailed information, analyses, and arguments. This abstract does not provide an accessible summary of the manuscript.

·         The keywords accurately reflect the content.

·         The introduction sets out the argument but is too general and does not summarize recent research related to the topic.

·         highlights gaps in current understanding or conflicts in current knowledge.

·         The results are clearly presented but poorly discussed. I strongly recommend including previous studies performed in other countries, and to deeper discuss the results.

·         Conclusions consider the limits of the study, and that is great, however, I would recommend the authors propose specific and effective interventions to improve physical activity in the pediatric oncology setting.

·         References are balanced, updated, and complete.

·         Appendix figures are clearly organized and well described.

Additional Remarks

-          Lines 40-42 – please, provide some references

-          Lines 77-78 – How do the authors explain the fact that males were 45 and females were only 23?

-           Did the authors exclude patients with previous psychiatric symptoms like anxiety and depression, eating disorders, OCD, ADHD… etc.? If so, this should be clearly specified. If not, this should be justified.

-          Why did the authors not report the BMI of the patients, % fat-free mass…)?

-          The statistical analysis paragraph is not clear and there is a problem in Lines 135-136. What about post-hoc analyses? What about inter-group differences?

Extensive editing required

Author Response

Thank you very much for reviewing our article. All your comments were very helpful for us and provided added value to make the text better. Please find the answers and our comments below:

  1. We have supplemented Simple Summary subsection (in lines 13-23) with results of our study: “Simple Summary: Physical and cardiorespiratory fitness deficits acquired during childhood often persist throughout adult life. It is crucial to understand the reasons for their development during childhood cancer and to prevent them from the early treatment stages. Children treated for malignancy exhibit a reduced level of physical activity (PA), but it varies depending on their place of residence. In conducted studies, the values of examined parameters were significantly lower in the group of Polish children treated for cancer compared to Czech children. It was also observed that a lower level of PA affected patients mental state. Children undergoing cancer treatment who declared a sedentary behaviour rated their health worse, while children engaging in more frequent PA (lasting at least 60 minutes daily), felt better mentally, reported higher physical performance, experienced fewer symptoms of depression. Adequate PA levels have an impact on improving physical fitness, the mental well-being, and preventing civilization diseases.”
  2. Abstract (in lines 24-37) has been completely corrected. We have added the summary of the results of our study. Now the Abstract subsection allows readers to get the gist or essence of the article quickly, preparing them to follow the detailed information, analyses, and arguments: “Abstract: This study aimed to assess the level of physical activity (PA) and quality of life of cancer treated children, depending on their place of residence (Poland vs the Czech Republic, where incidence and mortality rates of childhood malignancies are similar). 68 school-age children (7–18 years) undergoing oncological treatment were included in the study. The study used the quality of life questionnaire: KIDSCREEN-10 and the HBSC questionnaire. The study showed statistically significant differences in the level of PA between Polish and Czech children. In Poland, 93.75% of children exhibited no weekly physical effort at the level of moderate to vigorous PA. In the Czech Republic, 69.44% of children engaged in PA lasting at least 60 minutes per day, at least 1 day weekly. Physically active children engaging in more frequent effort, at least 60 minutes daily, re-ported higher physical performance (rho=0.41), higher energy levels (rho=0.41), less mood disturbance (rho=-0.31). Children with good relationships with parents were more likely to engage in submaximal PA and spent less time on stationary games. Our study showed that an appropriate level of PA improves the well- being and quality of life. It is crucial to promote attractive PA pro-grams tailored for cancer treated children.”
  3. Introduction subsection has been reorganized, and we have also added some valuable references. They address recent research related to this topic. Added articles and references are: 1) Sung et al. [1], Howlader et al. [2] – positions confirming the statistical data on cancer incidence in children, Linder et al. [10] – a position indicating that in a group of ill children the increase in sedentary habits is observed, Roussenq et al. [12] – children treated for cancer should undertake physical activity and should not be deprived of an opportunity to move. Find in in lines 79-82: “Children undergoing treatment for cancer should also engage in physical activity and should not be deprived of opportunities for exercise. Unfortunately, there are no adequate recommendations regarding the level of PA and sedentary behaviour for chronically ill and treated children [12].”

We also added relevant references (Wu et al. [13] and Ho et al. [14] regarding increased levels of fatigue, lack of physical inactivity and deterioration of quality of life in the group of children undergoing oncological treatment. Find it in lines 79-88: “Observations made by other researchers confirm that children undergoing treatment for cancer exhibit higher levels of fatigue, physical inactivity, and a decline in their quality of life [13]. In a study by Ho at al., researchers observed that an ade-quate level of PA during cancer is an important factor affecting the determinants of quality of life in a group of treated patients. In contrast, deficits acquired in childhood also persist long after cancer treatment [14].”

  1. In the Discussion subsection we have addressed the comments on our results and we have shown that also other researchers are interested in this topic, but there are no guidelines for children treated for cancer. We have also added a comment (in lines 390-407) on results of researchers from other countries:

“Unfortunately, there are no established standards defining the appropriate level of physical activity for children undergoing cancer treatment. Applying existing norms, which are proper only for healthy children, in the population of cancer treated children is unreliable. It is necessary to develop specific recommendations for this patients group.

In our study, it was observed that all children from Poland did not meet the recommendations for the minimum level of physical activity according to the MVPA indicator recommended for children. In the case of children from the Czech Republic, 25.01% of them engaged in such moderate activity for 5 or more days a week. Moreover, 93.75% of children from Poland did not engage in physical activity lasting at least 60 minutes daily for at least 1 day a week, while 69.44% of children from the Czech Re-public engaged in continuous physical activity for 60 minutes at least 1 day a week.

Studies conducted in other countries showed similar results. A prevalence of a sedentary lifestyle and reduced levels of physical activity were observed. Notably, this population remains inactive during treatment, immediately after its completion, and even many years after recovering from the disease. So far, WHO, ACSM (The American College of Sports Medicine), and CDC (Centers for Disease Control and Prevention)  have not developed specific recommendations for this population [12].”We also mentioned it in the Introduction section to prepare the reader for the topic of the article, and then expand on it in the results and discuss it more in the discussion section. In the Introduction section you will find it in lines 68-77: “An international study conducted in 25 countries (including the Czech Republic and Poland) found that the Czech children exhibited the highest levels of undertaken PA. At least 60% of Czech children spent > 2 hours a day on active play. By comparison, in Poland, less than 25% of children devoted that much time to active play. In the Czech Republic, nearly all children (98%) were actively spending time - at least for 1 hour daily, and 60% of children - over 2 hours daily. In Poland, the respective figures are 82% and 25%. On the other hand, way to school requiring physical activity (on foot or by bicycle) is reported by 44.5% of young Czechs and 39% of young Poles. A comparable number of children in the Czech Republic and Poland spend less than 2 hours a day in front of a screen (64% of children and adolescents in the Czech Republic and 59% in Poland, respectively) [11].”

  1. We have supplemented the Conclusion subsection. We have also proposed specific and effective interventions that can improve the PA level in a group of children treated for cancer. You can find it in lines 480-500: “Conclusions: Children treated for cancer exhibit reduced levels of physical activity. PA level is different depending on children place of residence. The values of the tested PA level were significantly lower in the group of children from Poland. In our study, it was observed that all children from Poland did not meet the recommendations for the mini-mum level of physical activity according to the MVPA indicator recommended for children. In the case of children from the Czech Republic, 25.01% of them engaged in such moderate activity for 5 or more days a week. Along with the decrease in the physical activity level, there is a decrease in the quality of life. Children declaring a sedentary lifestyle and, at the same time, limited contact with their peers assessed their health worse. Contacts with peers depended on the children's place of residence as well. Polish children reported significantly less time spent on playing and interacting with their peers.

We observed that unfortunately there are no defined standards of PA level for children treated for cancer. Applying existing norms, which are proper only for healthy children, in the population of cancer treated children is unreliable. It is necessary to develop specific recommendations for this patients group.

It is of utmost importance for health professionals to identify strategies that can help children and adolescents with cancer to increase undertaken physical effort and maintain regular activity. This is especially important during the active treatment pro-cess. Forms of physical activity that are attractive to current generation of children, who are accustomed to modern technology, should be sought. Interactive video games or programs using virtual reality could be appropriate.”

Additional remarks

  1. We have supplemented the references in Introduction – two of them are about the statistical data on cancer incidence in children:

1) Sung, H.; Ferlay, J.; Siegel, R.L.; Laversanne, M.; Soerjomataram, I.; Jemal, A.; Bray, F. Global Cancer Statistics 2020: GLOBOCAN Estimates of Incidence and Mortality Worldwide for 36 Cancers in 185 Countries. CA Cancer J Clin. 2021, 71, 209-249.

2) Howlader, N.; Noone, A.; Krapcho, M.; Miller, D.; Brest, A.; Yu, M. et al. SEER cancer statistics review, 1975–2016. Natl Cancer Inst. 2019, 1:16.

  1. The distribution of patients gender (more boys in the study) is random. We examined children hospitalized in clinics, and did not have any influence on the gender distribution, because children were randomly selected for the study.
  2. We have supplemented the exclusion criteria in lines 126-127: “The exclusion criteria were: intellectual disability, previous psychiatric disease, eating disorders, severe health condition resulting in contact difficulties.”
  3. We did not provide patients’ BMI indices because BMI is not a valid index in a group of children. The only thing we could add was data from percentile charts, but it is increasingly considered outdated and no longer relevant. Our examinations aimed to be a quick review of the children health from two clinics; therefore, we did not measure the percentage of body fat. We used data from patient records and epicrisis.
  4. We have supplemented a subsection Statistical analysis. We have also clearly indicated which post hoc tests were used by us, and emphasized that the differentiating factor was the study group. You can find it in lines 159-174: “In order to assess the statistical significance of result differences between groups (Group I - children from Poland, Group II - children from the Czech Republic) and verify whether groups are comparable, a single-factor analysis was used (with the group as the differentiating factor), to test the differences between groups within a given variable. The post-hoc analysis were conducted - the Dunn’s test was used. No significant differences between the groups were detected.”

We have carried out an extensive language editing throughout the whole text. Thank you once again for your feedback and comments. After implementing them, our text has improved.

Reviewer 4 Report

You did not include any line numbers and therefore, I can provide specific comments based on each sentence. However, I have tried my best to provide feedback that you can use to improve the quality of the manuscript.

The Abstract is not written well and does not clearly describe the aims and outcomes of the study. At times the sentences are incomplete and lacking details. A better description of the study design is needed, and the results should be supported by data and statistical analysis information e.g., p values and/or effect sizes etc.

Materials and Methods

2.1 Participants and Recruitment:

How were these children recruited for this study?

Why was greater than 7 years an inclusion criterion? What was the upper age limit?

2.2. Research Methods:

What is the validity of using the KIDSCREEN and International Health Behaviour Questionnaire HBSC in this study population? How relevant were these tools considering your study population is not healthy?

2.4 Statistical analysis:

“Shapiro-Wilk test”

There is no mention of what groups were compared and who they are.

Need to provide ranges for the strength of associations for Spearman’s Rank test e.g., strong, moderate, weak, etc.

Results

3.2. Physical health and mental well-being:

It is not clear what the numbers reported represent. I cannot see any p values.

For Spearman’s Rank Correlation, the values reported should be rho = xx

Error with closing brackets shown below:

…..Instagram, etc.). Snapchat, the Internet usage, etc.).

3.3. Autonomy and independence, as well as the relationships with parents:

Correct the error below related to the decimal place of the second value.

“…..as well as on weekends (-0.25, -027)…”

Table 6. “…sedentary behaviour (HBSC)…”

There appear to be a lot of results that have been previously reported in a previous paper. Therefore, I question what new information is being added.

4. Discussion

What are the physical activity needs/requirements for the study population? You cannot use the guidelines based on a healthy population because it is not specific and potentially not realistic. Please explore this topic and integrate it into the Discussion.

7. Conclusions

The final comments are convoluted and need to be revised.

There are numerous grammatical errors in the Introduction and the structure of sentences is poor. A person native to English needs to review the manuscript to improve the quality of the writing.

Author Response

Thank you very much for reviewing our article. All your comments were very helpful for us and provided added value to make the text better. Please find the answers and our comments below:

  1. We have corrected the Abstract subsection. Now it includes the description of the goals and results of our study. Sentences are complete and provide details. Abstract includes an improved description of a study project, the results are supported by data and information from statistical analysis with p-values. You can find it in lines 24-37: “Abstract: This study aimed to assess the level of physical activity (PA) and quality of life of cancer treated children, depending on their place of residence (Poland vs the Czech Republic, where incidence and mortality rates of childhood malignancies are similar). 68 school-age children (7–18 years) undergoing oncological treatment were included in the study. The study used the quality of life questionnaire: KIDSCREEN-10 and the HBSC questionnaire. The study showed statistically significant differences in the level of PA between Polish and Czech children. In Poland, 93.75% of children exhibited no weekly physical effort at the level of moderate to vigorous PA. In the Czech Republic, 69.44% of children engaged in PA lasting at least 60 minutes per day, at least 1 day weekly. Physically active children engaging in more frequent effort, at least 60 minutes daily, re-ported higher physical performance (rho=0.41), higher energy levels (rho=0.41), less mood disturbance (rho=-0.31). Children with good relationships with parents were more likely to engage in submaximal PA and spent less time on stationary games. Our study showed that an appropriate level of PA improves the well- being and quality of life. It is crucial to promote attractive PA pro-grams tailored for cancer treated children.”
  2. Materials and Methods: Participants and recruitment

We have added information on how children were recruited to the study (using computer program). You can find it in lines 107-112: “Children from Wroclaw and Prague clinics were qualified for the project using a computer program that randomly selected children for the study.”

  1. We have added a description regarding the age of the respondents. These were school-age children, therefore the age range was 7-18 years. Age school (7-18 years) was an inclusion criteria. Therefore, the upper age limit was 18 years. You can find it in lines 103-104: “A total group of 68 school-age children (7–18 years, 45 boys and 23 girls) were included.” and in lines 123-127: “The inclusion and exclusion criteria were defined. Inclusion criteria were: diagnosis of malignancy or haematological disease, age between 7 and 18 years, inpatient treatment in the hospital ward, hospital stay > 7 days, and chemotherapy treatment administered. The exclusion criteria were: intellectual disability, previous psychiatric disease, eating disorders, severe health condition resulting in contact difficulties.”
  2. Research Methods

Indeed, it would be better to use research tools intended for ill children. We chose such questionnaires because they are suitable for children and allow for comparative studies with healthy children. We have already conducted such studies and compared the PA and quality of life of sick and healthy children to see how much the results differ. It is true that it would be better to compare our results with results of different ill children, and we did it in the Discussion subsection in lines 402-403 however, there are no norms and tests adequate for cancer treated children: “Studies conducted in other countries showed similar results. A prevalence of a sedentary lifestyle and reduced levels of physical activity were observed. Notably, this population remains inactive during treatment, immediately after its completion, and even many years after recovering from the disease. So far, WHO, ACSM (The American College of Sports Medicine), and CDC (Centers for Disease Control and Prevention)  have not developed specific recommendations for this population [12].”

We also mentioned it in the lines 390-394: “Unfortunately, there are no established standards defining the appropriate level of physical activity for children undergoing cancer treatment. Applying existing norms, which are proper only for healthy children, in the population of cancer treated children is unreliable. It is necessary to develop specific recommendations for this patients group.”

  1. In the subsection of Statistical analysis, we have once again added information about which groups were compared (in lines 162-165): “In order to assess the statistical significance of result differences between groups (Group I - children from Poland, Group II - children from the Czech Republic) and verify whether groups are comparable, a single-factor analysis was used (with the group as the differentiating factor), to test the differences between groups within a given variable.”

We have also provided the range of correlation strengths in the statistical description: “The strength of Spearman's rank correlation were determined as: weak correlation < 0.2, low correlation 0.2-0.4, moderate correlation 0.4-0.6, high correlation 0.6-0.8, very high correlation 0.8-0.9, almost complete correlation 0.9-1.0.”

It seems to us that this will be easier for the reader to find in the results, while adding these descriptive values in the text would make it more difficult to perceive.

  1. Subsection Results: Physical health and mental well-being

We have supplemented  the tables from Results subsection with p-values. We have also added values of Spearman rank correlation in the description of the results throughout the while text (rho=xx).

  1. We have corrected punctuation errors and added missing brackets, periods, and typos.
  2. In Strength and Limitations subsection we have added crucial information. We have explained which new results are now included in our study. You can find it in lines 456-459: “The novelty of our research is that in the fact that so far no one has examined the level of physical activity and quality of life of children undergoing cancer treatment from Poland and the Czech Republic. So far, our project has examined these values, but has not conducted comparative studies with other research centers.”
  3. Cleared the issue regarding application of PA norms for both healthy and ill population. We have made an extensive research on this topic and added relevant literature to the Discussion subsection: “Unfortunately, there are no established standards defining the appropriate level of physical activity for children undergoing cancer treatment. Applying existing norms, which are proper only for healthy children, in the population of cancer treated children is unreliable. It is necessary to develop specific recommendations for this patients group.

In our study, it was observed that all children from Poland did not meet the recommendations for the minimum level of physical activity according to the MVPA indicator recommended for children. In the case of children from the Czech Republic, 25.01% of them engaged in such moderate activity for 5 or more days a week. Moreover, 93.75% of children from Poland did not engage in physical activity lasting at least 60 minutes daily for at least 1 day a week, while 69.44% of children from the Czech Re-public engaged in continuous physical activity for 60 minutes at least 1 day a week.

Studies conducted in other countries showed similar results. A prevalence of a sedentary lifestyle and reduced levels of physical activity were observed. Notably, this population remains inactive during treatment, immediately after its completion, and even many years after recovering from the disease. So far, WHO, ACSM (The American College of Sports Medicine), and CDC (Centers for Disease Control and Prevention)  have not developed specific recommendations for this population [12].”

We have added some information in Conclusion subsection: “We observed that unfortunately there are no defined standards of PA level for children treated for cancer. Applying existing norms, which are proper only for healthy children, in the population of cancer treated children is unreliable. It is necessary to develop specific recommendations for this patients group.”

  1. We have corrected Conclusion subsection. You can find it in lines 480-500: “Children treated for cancer exhibit reduced levels of physical activity. PA level is different depending on children place of residence. The values of the tested PA level were significantly lower in the group of children from Poland. In our study, it was observed that all children from Poland did not meet the recommendations for the mini-mum level of physical activity according to the MVPA indicator recommended for children. In the case of children from the Czech Republic, 25.01% of them engaged in such moderate activity for 5 or more days a week. Along with the decrease in the physical activity level, there is a decrease in the quality of life. Children declaring a sedentary lifestyle and, at the same time, limited contact with their peers assessed their health worse. Contacts with peers depended on the children's place of residence as well. Polish children reported significantly less time spent on playing and interacting with their peers. We observed that unfortunately there are no defined standards of PA level for children treated for cancer. Applying existing norms, which are proper only for healthy children, in the population of cancer treated children is unreliable. It is necessary to develop specific recommendations for this patients group. It is of utmost importance for health professionals to identify strategies that can help children and adolescents with cancer to increase undertaken physical effort and maintain regular activity. This is especially important during the active treatment process. Forms of physical activity that are attractive to current generation of children, who are accustomed to modern technology, should be sought. Interactive video games or programs using virtual reality could be appropriate.”

Thank you once again for your feedback and your valuable comments. After implementing them, our text has improved.

Reviewer 5 Report

1. Include the study design in the title.

2. What is the study recruitment period (from when to when)?

3. 68 is a very small sample size.

4. How the diagnosis of cancers was made?

5. How sample size calculation was done? Did the study have enough power?

6. What was the non-response rate at the time of recruitment?

7. Cite a reference for the KIDSCREEN-10 questionnaire and include the questionnaire in the appendix.

8. Cite a reference for the HSBC questionnaire. 

9. Explain in simple terms what is "single-factor analysis".

10. Since this is not a random sample, statistical tests should not be applied (correlation tests may be fine). Also, 68 sample likely does not have enough power to detect statistically significant differences between groups. Provide the results as descriptives or justify that the study had enough power. 

11. There are too many tables. Reduce the number of tables in the text and put the remaining in the appendix. 

12. Reduce the first paragraph of the conclusion. Summarize the key findings alone in this paragraph. 

My major concern is the very small sample size of the study. 

Author Response

Thank you very much for reviewing our article. All your comments were very helpful for us and provided added value to make the text better. Please find our answers and comments below:

  1. We included the project of the study in the title: “Physical activity level and quality of life of children treated for malignancy, depending on their place of residence: Poland vs the Czech Republic. An observational study.”
  2. In subsection 2. Materials and Methods 2.1. Participants and Recruitment, we precisely defined (in the line 102-103) the period during which the research was conducted: “The study included children undergoing oncological treatment (chemotherapy cycles in hospital settings) from 6th of March to 7th of April 2023.”
  3. Answer for the question number 3 and 5: We are aware that the sample could have been larger; however, we conducted calculations using the GPower The sample size was determined based on previous studies assessing the level of physical activity and quality of life in children with cancer, with an effect size of 0.6. It was determined that 76 patients (38 patients in each group) should be enrolled. We used GPower 3.1.9 software for this sample size calculation. The calculation was based on Spearman’s rank correlation, with a within-between interaction type I error rate set at 5% (α = 0.05), an effect size (q) of 0.6, and a type II error rate that provided 80% power for the two groups.

We wanted to conduct the research in a short period to assess children in the clinic simultaneously, under similar climatic and weather conditions - during the spring season. The results of the GPower test showed that study group could have been only a slightly larger.

We believe that this did not significantly affect the results and the power of the applied tests, the results of which are sufficiently robust. We think that the topic is important and the results are good enough to show them.

 Furthermore, the group of children with cancer is not a large group in general. Statistics, such as those for leukaemia, which is the most common cancer in childhood, confirm this fact. We are aware that the study group was relatively small. However, statistics on childhood cancer incidence in Poland confirm that the group of children I examined was actually substantial. In Poland, there are approximately 1100 to 1200 new cases of childhood cancer each year. Out of this group, around 360 children have leukaemia. The Lower Silesian voivodeship, where the Clinic for Bone Marrow Transplantation, Paediatric Oncology, and Haematology is located, has about 500,000 children. The incidence rate of leukaemia is 4 cases per 100,000 children. Analysing these data, we can conclude that in the Lower Silesian region, there are approximately 20 new cases of childhood leukaemia each year.

  1. In subsection 2. Materials and Methods 2.1. Participants and Recruitment, we included information in the text about how childhood cancers were diagnosed in the study group. You can find it in lines 109-112: “Cancer was diagnosed in case of every participant basing on a number of detailed examinations, including cytological evaluation, immunophenotype, genetic and molecular evaluation. Children were treated according to the appropriate protocol for given cancer type.”
  2. The answer you can find in point 3.
  3. At the recruitment stage of the study, 9 children did not complete the questionnaire entirely or did not submit it at all.
  4. In the "Research Methods" subsection, we provided references to the literature related to the Kidscreen questionnaire. You can find it in lines 130-132: “The study was based on the international quality of life questionnaire related to children’s and adolescents’ health – KIDSCREEN – 10 and the HBSC (Health Behaviour in School - aged Children) questionnaire [17].”
  5. In the "Research Methods" subsection, we provided references to the literature related to the HSBC questionnaire. You can find it in lines 143-144: “The physical activity level was assessed using the International Health Behaviour Questionnaire HBSC (Health Behaviour in School-aged Children) [18].”
  6. In the Statistics analysis subsection we explained what single-factor analysis is. You can find it in lines 162-165: “In order to assess the statistical significance of result differences between groups (Group I - children from Poland, Group II - children from the Czech Republic) and verify whether groups are comparable, a single-factor analysis was used (with the group as the differentiating factor), to test the differences between groups within a given variable.”
  7. We presented the results in descriptive form by providing the ranges of correlation strengths in the statistical description. Additionally, we calculated G*Power before the study and observed the group size, as described above.

Moreover in the Statistic analysis subsection, we have added a description of the correlation forces. It is in lines 171-173: “The strength of Spearman's rank correlation were determined as: weak correlation < 0.2, low correlation 0.2-0.4, moderate correlation 0.4-0.6, high correlation 0.6-0.8, very high correlation 0.8-0.9, almost complete correlation 0.9-1.0.”

  1. In the guidelines of the journal Cancer, there are no limits on tables or words in the text. It seems to us that reducing the number of tables may negatively impact the presentation of the results. We have also now added p-values to the tables.
  2. We shortened the first paragraph of the Conclusion and reorganized this subsection. We also included certain information required by the other reviewers. Now, this subsection is more comprehensive and better reflects the key results and conclusions, while also presents solutions for this group of patients.

We have carried out an extensive language editing throughout the whole text. Thank you once again for your feedback and comments. After implementing them, our text has improved.

Round 2

Reviewer 3 Report

The authors provided an extensive revised version of the manuscript.

English has been improved

Reviewer 4 Report

Well done on addressing my comments and improving the quality of your manuscript.

Some very minor grammatical errors.

Reviewer 5 Report

Thanks for addressing my concerns. 

Fine.